# Revised Hammersmith Scale for spinal muscular atrophy: Inter and intra-rater reliability and agreement

**Danielle Ramsey**[1¤]*, **Gita Ramdharry**[2], **Mariacristina Scoto**[1], **Francesco Muntoni**[1,3], **Amanda Wallace**[2,4], **on behalf of the SMA REACH UK network**[¶]

**1** Dubowitz Neuromuscular Centre, UCL Great Ormond Street Institute of Child Health, University College London, London, United Kingdom, **2** Queen Square Centre for Neuromuscular Diseases/UCL Department of Neuromuscular Diseases, University College London, London, United Kingdom, **3** National Institute for Health Research Great Ormond Street Hospital Biomedical Research Centre, UCL Great Ormond Street Institute of Child Health, London, United Kingdom, **4** UCL Great Ormond Street Institute of Child Health, University College London, London, United Kingdom

¤ Current address: School of Health and Sports Sciences, University of Suffolk, Ipswich, Suffolk, United Kingdom

¶ Membership of the SMA REACH UK network is provided in the Acknowledgments.

* danielle.ramsey.12@ucl.ac.uk

**Data Availability Statement:** The minimal dataset underpinning the results of this study can be found in the Supporting Information S1 Dataset. This

## Abstract

The Revised Hammersmith Scale (RHS) for Spinal Muscular Atrophy (SMA) was designed as a psychometrically robust clinical outcome assessment to assess physical abilities of patients with type 2 and 3 SMA. The reliability properties of the RHS have not yet been reported. A prospective RHS reliability study was undertaken in a UK cohort of experienced neuromuscular paediatric Physiotherapists. Reliability testing was conducted via a virtual survey platform two weeks apart. Through the virtual platform participants scored videos of two RHS assessments, one of a child with SMA 2 and one of a child with SMA 3. Inter and intra-rater reliability was analysed using a type 3 Intraclass Correlation Coefficient (ICC). Intra-rater agreement was further analysed using Bland Altman (BA) Limits of Agreement (LOA) and plots. The acceptable inter and intra-rater variability was set as a change of ± 2 by the international team of expert physiotherapists who developed the RHS. Inter-rater agreement, n = 22 raters, type 3 ICC was 0.989 (95% CI 0.944 to 1.00), 97.7% of scores were within the acceptable limits of ± 2 points. Intra-rater agreement, n = 21 raters, type 3 ICC ranged from 0.922 to 1.0, with 97.6% of scores within the acceptable limits of ± 2 points. The mean SMA 2 intra-rater difference was -0.10 (-0.6 to 0.4), with lower LOA -2.24 and upper LOA +2.04. Intra-rater difference between tests for SMA 3 intra-rater difference was -0.05 (-0.6 to 0.5), with lower LOA -2.48 and upper LOA +2.38. Intra-rater scoring precision fell within BA agreement limits of ±2 points. The results demonstrate that the RHS is highly reliable when used by experienced UK physiotherapists, and variability of test scores regarding inter and intra-rater reliability was confirmed to lie within ±2 points.

does not include detail regarding the demographics of the raters included in the study, this information, their affiliation with the SMA REACH UK sites and small population studied would mean even anonymised data would have several indirect identifiers which may risk breaking anonymity. Minimal information pertaining to the raters can be found within the participants section within results. The minimal dataset can also be found within the results section in tables 1, 2 and 3 and within Figures 1, 2 and 3. Data beyond the minimal dataset required for this study are not publicly accessible due to it containing sensitive data pertaining to human research participants and may compromise anonymity. However, external parties can request data in aggregate form from the SMA REACH UK steering committee for researchers who meet the criteria for access to confidential data. Please contact Dr Salma Samsuddin, SMA REACH UK & ISMAC UK Trial Manager via salma. samsuddin@gosh.nhs.uk for any queries regarding data access.

**Funding:** This study was supported, in the UK, by the SMA REACH UK project (https://eur03. safelinks.protection.outlook.com/?url=http%3A% 2F%2Fwww.smareachuk.org%2F&data=05% 7C01%7Cd.ramsey%40uos.ac.uk%7C98a052c30 94e439151f408dad3bba1fb%7Cee265dd904ad41 b7b409e6699705d35d%7C0%7C0%7C63805510 1504580676%7CUnknown%7CTWFpbGZsb3d8e yJWIjoiMC4wLjAwMDAiLCJQIjoiV2luMzIiLCJBTiI 6Ik1haWwiLCJXVCI6Mn0%3D%7C3000%7C%7C %7C&sdata=NbBTTjSSCi255fq%2BahVKpkKOoJQ n5h%2BGUC%2FoX6%2FTYVE%3D&reserved= 0). FM is the chief investigator of the SMA Reach project. Commercial funding for the SMA REACH UK project is provided by Biogen Inc (REC reference: 13/LO/1748, IRAS project ID: 122521), via UCL and GOSH. Historically, funding of the SMA REACH UK Project has also been provided by the SMA Trust and Muscular Dystrophy UK (07DN02; 37787 https://eur03.safelinks.protection. outlook.com/?url=http%3A%2F%2Fwww. musculardystrophyuk.org%2Fgrants%2Fclinical-trial-coordinators%2F&data=05%7C01%7Cd. ramsey%40uos.ac.uk%7C98a052c3094e43915 1f408dad3bba1fb%7Cee265dd904ad41b7b409e6 699705d35d%7C0%7C0%7C63805510150458806 76%7CUnknown%7CTWFpbGZsb3d8eyJWIjoiMC 4wLjAwMDAiLCJQIjoiV2luMzIiLCJBTiI6Ik1haWwi LCJXVCI6Mn0%3D%7C3000%7C%7C%7C&s data=m5jiQW5nn8puWj1LU5QLJ2ePuM6hlsi05u 5lvcf2%2F04%3D&reserved=0), the MRC Translational Research Centre at UCL and Newcastle (MR/K501074/1), and the National Institute for Health Research Biomedical Research Centre (515048) at Great Ormond Street Hospital

## Introduction

Spinal Muscular Atrophy (SMA) is a neuromuscular condition characterised by biallelic mutations of the Survival Motor Neuron 1 (*SMN1*) gene [1]. The absence of *SMN1* adversely affects the integrity of the anterior horn cell in the spinal cord leading to degeneration of alpha motor neurons and subsequent muscular atrophy, resulting in a varying clinical phenotype of SMA [2–4]. In the severest forms of SMA, type 0 and 1, patients will never achieve the ability to sit and survival ranges from the first few days or weeks of life to less than two years [4]. In types 2, 3 and 4 SMA survival into adulthood is expected and physical presentation differs with sitting being the highest achieved physical ability for type 2 and walking the highest ability achieved in type 3 and 4 [4, 5].

The first targeted treatment for SMA, Nusinersen, was licensed by both the Food and Drug Administration (USA), in December 2016, and the European Medicines Agency, in June 2017 [6–8]. Several other potential therapeutics are also under investigation, or have received recent approval, such as risdiplam and onsaemnogene abeparvovec [3, 6, 7, 9–11]. Functional scales are key clinical outcome measurement tools used to monitor SMA both in the clinical setting and to measure efficacy of therapeutics being tested in clinical trials [1, 12–16]. The rapid progression of the field, promising early signs of therapeutics, and demands from regulatory authorities means there is greater need for scales which not only measure the disease-specific nature of this condition but also have the capacity to demonstrate potential improvement not seen before in the natural history of SMA.

The Revised Hammersmith Scale for SMA (RHS) was developed as a 'next generation' SMA specific scale to meet the requirements of today's climate namely psychometrically robust, grounded in clinical sensibility and with the capacity to capture improvement in patients with type 2 and 3 SMA and the evolving treated phenotypes [17]. A large international pilot demonstrated the RHS was able to capture a broad spectrum of ability across SMA types 2 and 3, it was able to distinguish between clinically different groups, and although a small floor effect (n = 1) was noted it has no ceiling effect [17]. The International Rare Diseases Research Consortium Task Force (IRDiRC) on Patient Centred Outcome Measures (PCOMs) recommend Rasch Measurement Theory (RMT) methodology for the design of sophisticated and psychometrically robust PCOMs, and state that a distinct benefit of the RMT approach is the ability to detect treatment benefit [18]. The IRDiRC recently highlighted the Revised Hammersmith Scale for SMA (RHS) as a good example of the use of RMT [18]. The RHS is cited in the National Institute for Clinical Excellence (NICE) managed access agreement for Nusinersen as an endpoint for measuring treatment efficacy in patients with type 2 and 3 SMA [19, 20]. Furthermore the RHS is also being used in clinical trials to measure treatment efficacy [21]. Inter-rater reliability and intra-rater reliability have not yet been investigated and therefore reliability and agreement of this measurement tool is not documented. This study aimed to investigate and describe the inter and intra-rater reliability of the RHS when used by experienced neuromuscular physiotherapists in the UK for the assessment of patients with SMA type 2 and 3.

## Methods

### Reliability study design overview

A prospective reliability study was conducted, via a virtual platform, in a UK cohort of paediatric physiotherapists (raters) with experience in neuromuscular diseases. The raters viewed videos of an RHS assessment undertaken by the investigating physiotherapist on two patients with SMA: one with type 2 SMA and one with type 3 SMA. These videos were viewed and

for Children NHS Foundation Trust and University College London (https://eur03.safelinks.protection. outlook.com/?url=http%3A%2F%2Fwww.gosh. nhs.uk%2Fresearch-and-innovation%2Fnihr-great-ormond-street-brc%2Fabout-brc&data=05% 7C01%7Cd.ramsey%40uos.ac.uk%7C98a052c30 94e439151f408dad3bba1fb%7Cee265dd904ad41 b7b409e6699705d35d%7C0%7C0%7C6380551 01504580676%7CUnknown%7CTWFpbGZsb3d8 eyJWIjoiMC4wLjAwMDAiLCJQIjoiV2luMzIiLCJBTi l6Ik1haWwiLCJXVCI6Mn0%3D%7C3000%7C% 7C%7C&sdata=hP8CekkkaxQFQVJ6QOu0fFJYaW Rhn0NxV0JRADs3sGU%3D&reserved=0). The funders had no role in study design, data collection and analysis, decision to publish, or preparation of the manuscript.

**Competing interests:** I have read the journal's policy and the authors of this manuscript have the following competing interests: MS reports participation to Scientific Advisory boards and teaching initiatives for Avexis, Biogen, Roche; she is involved as an investigator in clinical trials from Avexis, Biogen and Roche FM reports participation to Scientific Advisory boards and teaching initiatives for Avexis, Biogen, Roche and Novartis. He is member of the Rare Disease Scientific Advisory Board for Pfizer. He is involved as an investigator in clinical trials from Avexis, Biogen and Roche. In addition he is the principal investigator of the SMA REACH UK clinical network, partially funded by Biogen and by SMA UK. DR reports participation to teaching initiatives for Roche. GR reports participation in consultancy for Orphazyme. AW reports no disclosures.

subsequently scored by the participating raters on two separate occasions via two secure online password protected surveys. This study is reported in keeping with the guidelines for reporting reliability and agreement studies (GRAAS) [22].

## Revised Hammersmith Scale

The RHS is a clinician rated SMA specific outcome measure containing 36 items which assess physical motor performance [17]. The scale assesses motor functional activities related to sitting, supine, rolling, prone, ability to move and get up from the floor, balance, standing, run/walk, stairs, ascending and descending a step and the ability to jump. Thirty-three items are graded according to an ordinal 0, 1, 2 scale where 0 represents the least physical ability or function achieved, and 2 the highest. Three items are graded 0 and 1 where 0 represents an inability to complete the item, and 1 represents achieving the item. Two timed tests are included within the scale, and WHO motor milestones can also be completed concurrently. The scale was developed using modern psychometric techniques and latent measurement theory via the Rasch Unidimensional Measurement Model (unrestricted and simple logistic model) [17, 18]. Rasch analysis identified unidimensionality of the RHS as acceptable with t-test 7.3%, binomial test lower 95% confidence interval proportion, 0.05 [17]. Reliability of the RHS was demonstrated to be good with a high Person Separation Index (PSI) of 0.98 [17]. Dependency was seen between items tested on the right and left and rolling supine to prone and prone to supine however removing these items did not alter the PSI [17].

## RHS training

Twenty-seven physiotherapists from 13 UK sites received training on the RHS at the North Star Network/SMA REACH UK meeting on 22nd April 2015. This is an annual meeting for North Star and SMA REACH UK centres/networks (https://www.northstardmd.com/ and http://www.smareachuk.org/) attended by neuromuscular physiotherapists who are involved in the care of patients with SMA and Duchenne Muscular Dystrophy. Additionally, physiotherapists at the SMA REACH UK sites (which in 2015 was London and Newcastle) who were unable to attend the meeting in April received direct training from the lead SMA REACH UK physiotherapists (DR, AM). Training consisted of provision of an RHS Manual (version 1 21.04.2015), and RHS testing proformas (version 17.03.2015). These documents correspond with the final published version of the RHS in 2017 [17]. There was detailed and comprehensive discussion with demonstration on how to test and score each RHS item. The physiotherapists had opportunities to ask questions throughout the training.

## Raters

All UK physiotherapists trained in the use of the RHS were invited to participate in this study. As this scale had not been published at the time of conducting the reliability study the population of raters were the only ones in the UK who were trained in use of the RHS. As a result, the sample of raters invited to participate was representative of the whole population.

The inclusion criteria for the study were: all participants must have attended RHS training; have at least one SMA patient on their current clinical caseload; been a qualified physiotherapist for at least two years; in current employment as a physiotherapist; given their informed consent to participate; have completed and returned a non-disclosure agreement (required for viewing the testing videos via the virtual platform). Only participants who completed survey one (inter-rater testing) were invited to complete survey two (intra-rater testing).

The minimum number of participants was set as 20 for this study. This was calculated using a sample size estimator for the Bland Altman Limits of Agreement, assuming standard deviation of the repeats would be 2, precision would be 1.55 [23].

## Reliability testing protocol

Reliability testing was conducted by two identical online surveys, where participating raters scored video clips of two SMA patients being assessed by the investigating physiotherapist (DR).

**Survey design.**   Reliability testing was conducted via two online surveys, using UCL Opinio 7 survey platform [24]. All questions in the survey were mandatory and were designed with checks in place to ensure their completion. In each survey raters viewed item by item video clips of the RHS assessment of two patients, one with SMA type 2, and one with SMA type 3 who were enrolled on the SMA REACH UK study and who gave their explicit informed consent to participate as models for the reliability study. Each question on the survey was an item of the RHS and consisted of a video clip of that item being assessed, an online proforma of the RHS descriptors for that item [17], and a box to indicate their score for that item. Raters were permitted to use the RHS manual version 1.0 21.04.2015 when scoring the item to replicate good clinical practice when testing an item.

Inter-rater reliability was investigated with survey one (S1) and intra-rater reliability was assessed two weeks later in survey two (S2). S1 contained additional information pertaining to professional experience and experience of using relevant neuromuscular outcome measures. Participants completed two RHS assessments (one SMA 2 and one SMA 3) per survey. The same two assessments were then viewed and scored again in S2, two weeks later. Two weeks were chosen to ensure enough time between surveys so that the participants remained blind to their previous scores and to maintain currency and engagement with the study. Each survey took approximately 45mins to 1 hour to complete with the study occupying approximately 2 hours of the participants time in total.

Reliability testing occurred no sooner than one month following RHS training. This allowed participants to familiarise themselves with the RHS in the clinical setting prior to undertaking the study.

Each survey had a start and stop date to control and restrict responses within a specified time frame and was open for 3 days allowing participants a degree of flexibility to choose a convenient time to complete the survey. During survey completion raters could not go back and change their scores, and after completion they no longer had access to the survey ensuring they remained blind to their previous results.

**Patient videos.**   The RHS assessment videos used in this study were taken using a Go Pro Hero 3 White Edition (assessments were undertaken and videoed by DR). Videos were stored in accordance with NHS information governance guidelines to meet the standards of the NHS Information Governance toolkit [25]. The videos were edited using Windows Movie Maker (version 2012). Ethical approval via the SMA REACH UK project was granted for the collection and use of these videos in this study (London–Bromley REC reference 13/LO/1748) and informed consent was obtained for the participation in and recording of the assessment videos (this included parental/guardian informed consent together with minors giving their informed assent).

The reliability testing surveys were designed to minimise any risks of accidental or deliberate disclosure of assessment footage. The resultant protocol was approved by the SLMS UCL Information Services Division Information Governance Lead, Caldicott Guardian and Deputy Medical Director at Great Ormond Street Hospital. Each survey was password protected, and

participants received a unique survey URL link, username and key. Each unique URL could only be used once.

## Statistical analysis

The RHS is an ordinal scale which produces an overall total numeric score. The numeric total score was used to analyse reliability. The level of agreement of the RHS total scores between raters and intra-rater was also used to determine reliability.

The type 3 Intra-class Correlation Co-efficient (ICC), two way mixed for absolute agreement model (single measures), was chosen for both inter and intra-rater analysis, this was due to the fixed population of raters studied, and absolute agreement was chosen to investigate systematic error [26]. The level of agreement between/within rater scoring, the degree to which scores differed, was also investigated [27]. The study design ensured consistency of patient assessment via a single videoed assessment of each patient which the raters then scored twice 2 weeks apart, patient change over time or performance in repeated assessments was therefore not a factor in this study. This study focussed solely on the precision and reliability of the physiotherapist raters scoring of these assessment videos. To investigate the level of agreement (precision) of scale scoring expert physiotherapists were consulted regarding setting the level of agreement for the Bland Altman analysis [17]. Evidenced values for the level of agreement were unavailable at the time of this study due to the RHS being a new scale and entering a pilot phase of testing, therefore properties such as scale variance had not yet been investigated. The experts agreed that the limits of agreement (precision) of the anticipated differences between raters/intra-rater for the total score of the test would lie within ±2 points overall and therefore ±2 was the set limits of agreement (precision) for this study. This level was based upon their expert experience of using the Hammersmith Functional Motor Scale (HFMS), Hammersmith Functional Motor Scale Expanded (HFMSE) and their involvement in developing and testing the RHS. The levels of agreement set by the experts, at the time of this study, were not dissimilar to the scale variance reported in the literature for the HFMS, HFMSE, and Modified HFMS [28–31]. Descriptive statistics were used to interpret the level of agreement.

The Bland Altman (BA) Limits of Agreement (LOA) analysis was conducted to investigate intra-rater reliability with two replicate observations and multiple raters. This test was chosen as it is more grounded in the data, easier to interpret and clinically useful as it analyses the magnitude of measurement in addition to the agreement [32–34]. Data was presented in the form of descriptive statistics: mean intra-rater difference and 95% CI, upper and lower LOA with 95% CI and BA plots. The pre-set limits of agreement for this test remained at an acceptable difference of ±2 points. Rater demographics were analysed using descriptive statistics.

## Ethical approval

The study protocol was given a favourable ethical opinion by the UCL Research Ethics Committee (REC) on 11/05/2015 REC reference 6639/001 subsequent amendments were approved by the committee on 08/06/2015 and 14/08/2015. Research & Development approval was granted from Great Ormond Street Hospital joint research office to conduct this study with NHS staff. Local research and development approval from NSCN NHS sites was sought to include their staff in this study. All raters participating in the study gave their written informed consent to participate.

This study was affiliated to the longitudinal observational cohort study SMA REACH UK. SMA REACH UK was granted ethical approval (London–Bromley REC reference 13/LO/1748) to record assessment videos of participants, whom had given their informed consent for training purposes (for minors this included parents/guardians informed consent together with

minors giving their informed assent). To use these videos in this reliability study a substantial amendment for SMA REACH UK 13/LO/1748 was submitted to the Bromley NHS REC on 28/01/2015 and was granted favourable opinion on 05/02/2015.

## Results

When interpreting results presented for both inter and intra-rater reliability it is important to note that the RHS is scored in whole numbers, it is not possible to achieve a decimalised score. In order to allow for more in-depth understanding/analysis of this study decimalised scores are presented. However, regarding clinical meaningfulness/interpretation of the values these should be rounded up or down to a complete whole number as this would reflect the how the RHS would be scored clinically.

### Participants

Twenty-two Physiotherapists gave their informed consent and were included as participants (raters) in this study. Twenty-one physiotherapists met the full inclusion criteria for participation and one physiotherapist contacted the investigator stating they met all but one of the inclusion criteria where they had not assessed an SMA patient in the last year. This participant had significant proven experience, 14 years, in the neuromuscular field. A check sub-analysis found that this participant was not a significant outlier from a clinical or statistical perspective and therefore this participant was included in this study.

A total of 22 participants completed S1 (inter-rater testing) and 21 participants completed S2 (intra-rater testing). The participants were all experienced physiotherapists with a minimum of 5 years post-qualification experience, median 15.25 years (IQR 10 to 26), and had at least one year of experience treating children with neuromuscular conditions, median 9.5 years (IQR 4 to 14). There was wide variability regarding the number of SMA patients seen by each participant in the last a year varying from 0 (n = 1) to 50, the distribution was positively skewed with median 11 SMA assessments in the last year (IQR 6 to 20) reflecting the difference in patient distribution across specialist centres in the UK.

The functional scales reported to be used routinely by the participants to assess patients with SMA were the North Star Ambulatory Assessment (45.5%) and the Hammersmith Functional Motor Scale (40.9%), the Revised Hammersmith Scale (RHS) was used routinely in 22.7% of participants and Hammersmith Functional Motor Scale Expanded in 18.2%. A very small number of participants, n = 2 (9.1%), stated they were not familiar with the Hammersmith Functional Motor Scale, Hammersmith Functional Motor Scale Expanded or North Star Ambulatory Assessment scales, however all were aware of the RHS following the training.

The surveys (S1 and S2) were completed by 81.8% of participants at 9–10 weeks following initial training (June-July 2015) and 18.2% (n = 4) of participants at 39 weeks following training. The four participants included in the second wave of testing (39 weeks post-training) involved three physiotherapist who were invited to participate in the original testing but were unavailable (n = 1) or unable due to increased work pressures (n = 2). With regards the final participant, the local trust research and development approval was received once the original study testing had already begun and so could not participate initially but was invited to the second round of testing. A sub-analysis, using Mann Whitney U Test, was conducted to investigate whether this discrepancy in timing of the investigation since training had any effect on the inter and intra-rater scoring and no clinical or statistical differences in scores were observed (inter-rater scoring SMA 2 p = 1.00, SMA 3 p = 0.081; intra-rater scoring SMA 2 p = 0.763, SMA 3 p = 0.120).

**Table 1. Inter-rater reliability n = 22 raters.**

| | RHS Total Score | | | % RHS total scores sitting within ± points of the mean | | | |
|---|---|---|---|---|---|---|---|
| | Mean (95% CI) | SD | Range of difference either side of mean | ± 3 | ± 2 | ± 1 | ± 0 |
| SMA 2 | 13.2 (12.8, 13.7) | 1.07 | -2 to +2 | 100 | 100 | 86.4 | 27.2 |
| SMA 3 | 41.5 (40.9, 42.0) | 1.14 | -3 to +2 | 100 | 95.5 | 77.3 | 36.4 |
| SMA 2 & 3 | - | - | -3 to +2 | 100 | 97.7 | 84.1 | 29.5 |

SD: Standard Deviation; CI: Confidence Intervals.

### Inter-rater reliability–survey 1 (n = 22)

The inter-rater reliability results are presented in Table 1 and in Fig 1. The mean RHS total score for SMA 2 was 13.2 (95% CI 12.8, 13.7) and for SMA 3 was 41.5 (40.9, 42.0). The inter-rater reliability ICC (type 3) was 0.989, (0.944 to 1.00) demonstrating a very good level of agreement between raters according to the categories described by Altman [35]. With regards the ± 2 expert defined limits of acceptable agreement, for both SMA 2 and SMA 3 the 95% confidence intervals for RHS scores sat within ± 1 point difference of the mean. Furthermore, when looking at the entire set of values 100% of SMA 2 scores sat within ± 2 points, compared with 95.5% of values in the SMA 3 test, demonstrating a high level of agreement between raters. Inter-rater reliability of the RHS total scores in survey 2 returned an ICC (type 3) value of 0.997 (0.984, 1.0), again confirming high inter-rater reliability for this scale.

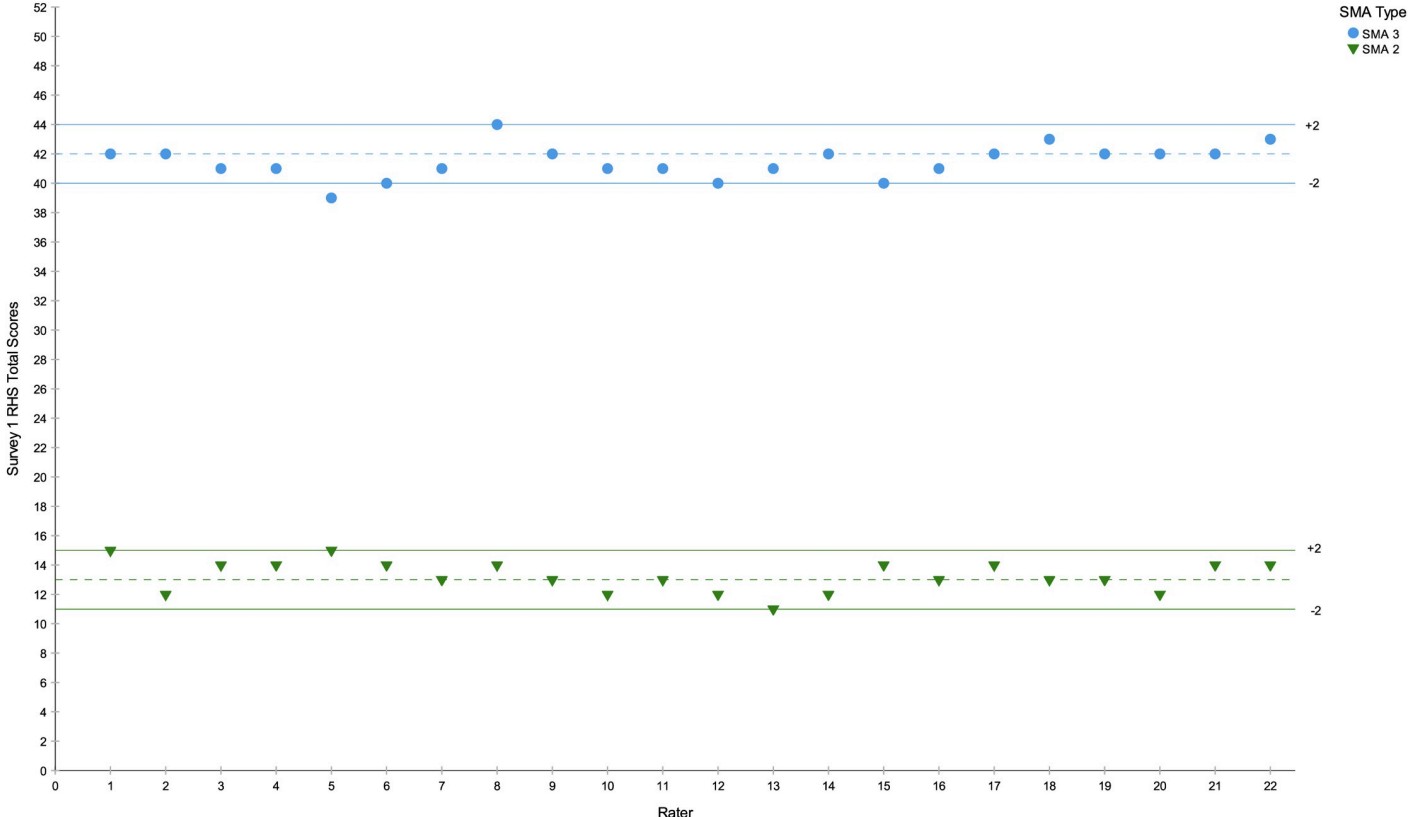

**Fig 1. SMA type 2 and 3 RHS inter-rater total scores with mean and ± 2 expert opinion of acceptable margin of error.**

**Table 2. Intra-rater reliability n = 21 raters.**

| | RHS Total Score | | Intra-rater difference | | | | % RHS difference scores sitting within ± points of the mean | | | |
|---|---|---|---|---|---|---|---|---|---|---|
| | Mean (95% CI) | SD | Mean (95% CI) | BA Lower LOA (95% CI) | BA Upper LOA (95% CI) | Range | ± 3 | ± 2 | ± 1 | ± 0 |
| SMA 2 | 13.3 (12.8, 13.8) | 1.15 | -0.10 (-0.6, 0.4) | -2.24 (-3.04, -1.44) | +2.04 (1.24, 2.84) | -2 to +3 | 100 | 95.2 | 85.7 | 52.4 |
| SMA 3 | 41.6 (41.1, 42.0) | 1.03 | 0.05 (-0.6, 0.5) | -2.48 (-3.4, -1.56) | +2.38 (1.46, 3.30) | -2 to +2 | 100 | 100 | 76.2 | 23.8 |
| SMA 2 & 3 | - | - | - | - | - | -2 to +3 | 100 | 97.6 | 81.0 | 38.1 |

BA: Bland Altman; LOA: Limits of agreement; CI: Confidence Intervals, SD: Standard Deviation.

Fig 1 highlights rater 5 as an outlier in the SMA 3 assessment with the greatest difference in score from the mean as -3, their SMA 2 assessment did however sit within + 2 of the mean.

## Intra-rater reliability–survey 2 (n = 21)

The intra-rater results are presented in Table 2. Intra-rater analysis in the form of Bland Altman plots are presented in Figs 2 and 3.

The mean RHS total score for SMA 2 was 13.3 (12.8, 13.8) and SMA 3 was 41.6 (41.1, 42.0), these raw scores are almost identical to the mean scores rated at the inter-rater testing in survey 1, with the 95% confidence interval within ±1 point of the mean for both SMA types. Intra-rater reliability for the 21 raters was found to be very high with ICC (type 3) values for individual raters ranging from 0.922 to 1.0, Table 3.

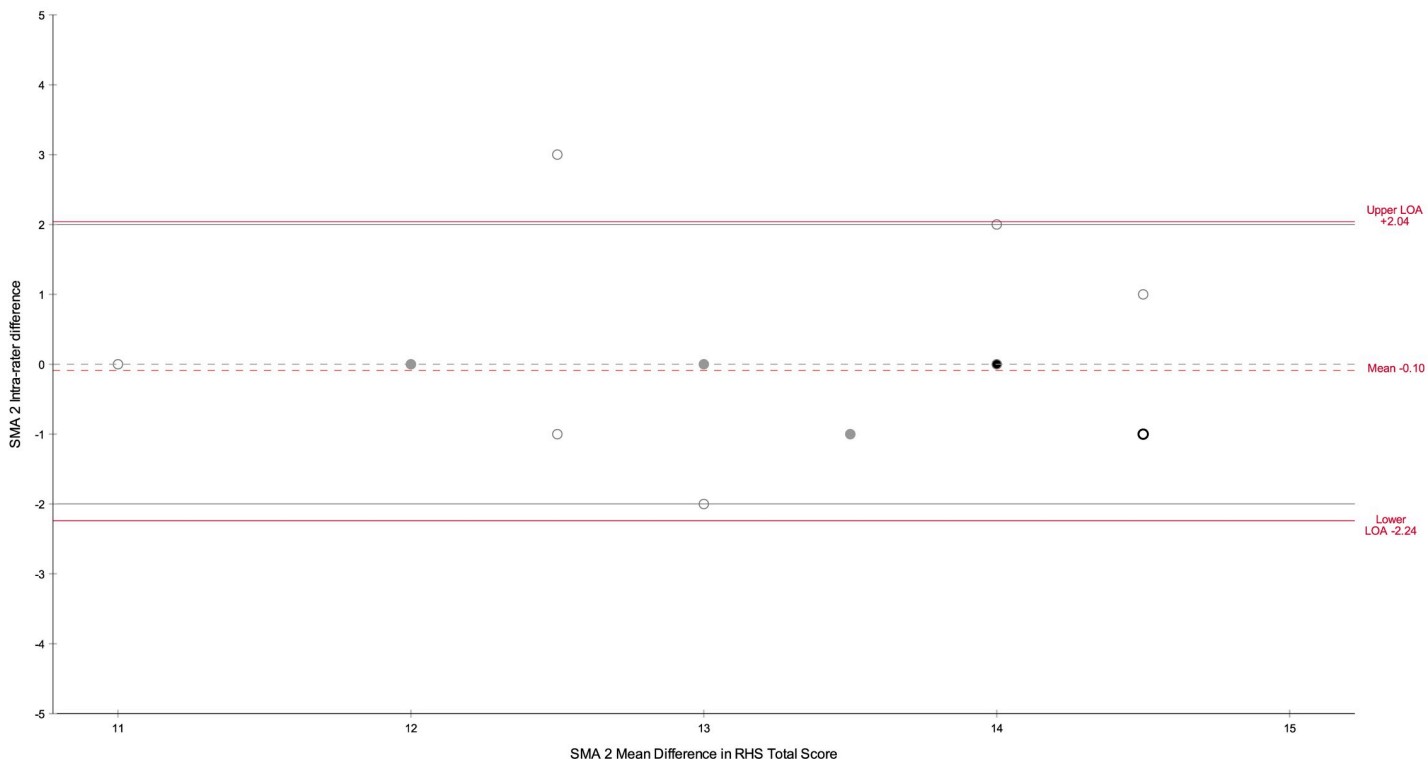

**Fig 2. SMA 2 Intra-rater Bland Altman plot.** Grey dotted line indicates 0 mean difference between tests, grey solid lines indicate the expert set limits of agreement. Dots–grey outline 1 rater, black outline 2 raters, grey filled dot 3 raters, black filled dot 4 raters.

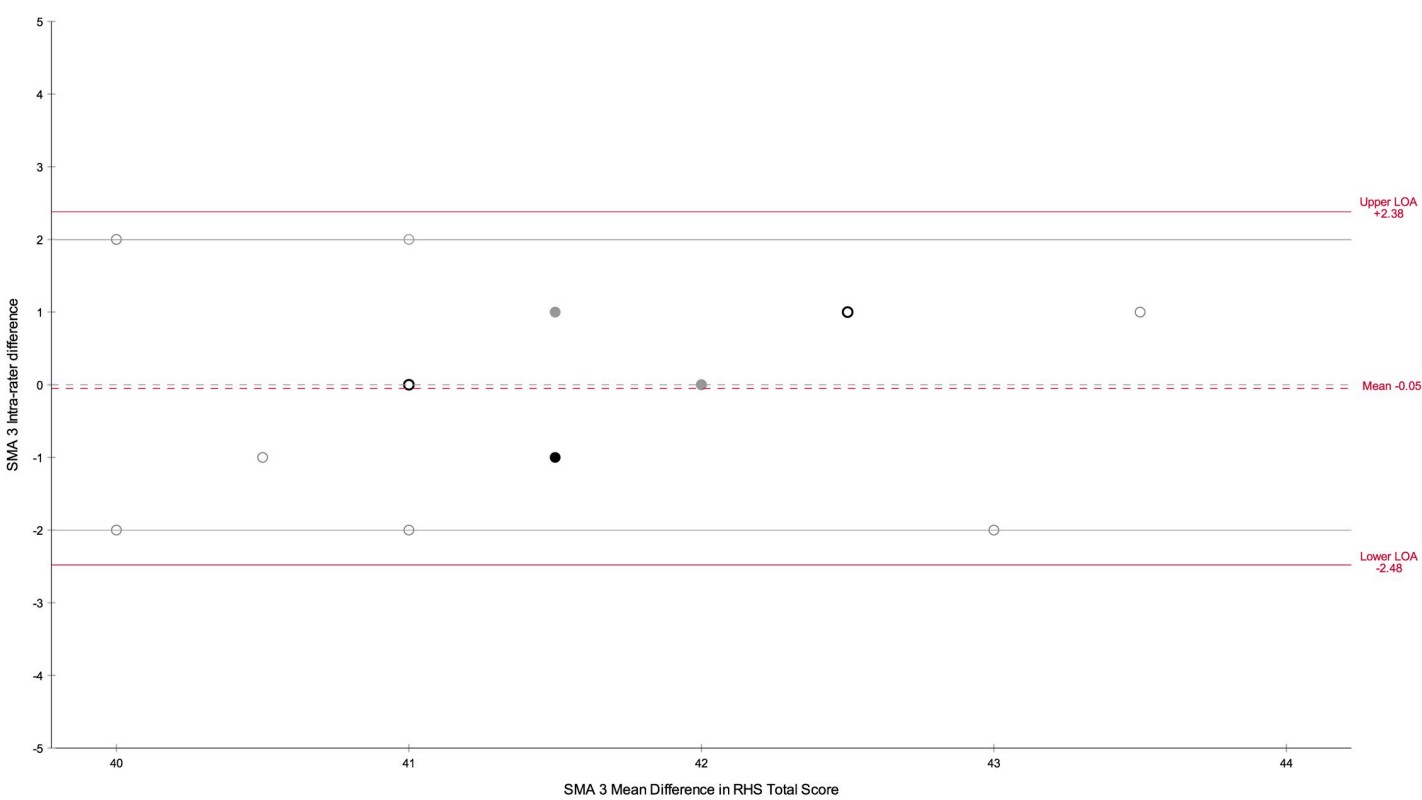

**Fig 3. SMA 3 Intra-rater Bland Altman plot.** Grey dotted line indicates 0 mean difference between tests, grey solid lines indicate the expert set limits of agreement. Dots–grey outline 1 rater, black outline 2 raters, grey filled dot 3 raters, black filled dot 4 raters.

Within pair differences regarding S1 and S2 RHS total scores were calculated for each rater. The mean intra-rater difference was observed to be -0.10 (-0.6, 0.4) for SMA 2 and SMA 3 0.05 (-0.6, 0.5), indicating a high confidence that there was no observable difference between testing scores for SMA 2 or SMA 3. The BA plots, Figs 2 and 3, show random scatter for both the SMA 2 and SMA 3 assessments indicating no systematic bias in the results. The BA LOA for SMA 2 were -2.24 to + 2.04, and SMA 3–2.48 to +2.38, see Table 2. When rounding to the whole score value, as would be seen clinically, both LOA's sat within the ± 2 set by the expert panel. The wide confidence intervals surrounding the upper and lower LOA may be indicative of a potential type 2 error due to small sample size, this is confirmed by 97.6% of actual values being within ± 2.

## Discussion

This study has for the first time investigated the reliability properties of the Revised Hammersmith Scale. National testing was conducted using a free and secure online survey system and was deemed to be a success following feedback from participants with a high response rate from both inter and intra-rater testing, 85.7% and 94.4% respectively. The protocol employed for this study could easily be replicated for both national and international training, and the format of reliability testing via video item analysis is similar to that used to test clinical evaluator reliability in clinical trials [28, 36, 37]. Video analysis via a virtual platform is useful for establishing inter/intra observer agreement and quality with regards scoring the items, but

**Table 3. Intra-rater type 3 ICC values.**

| Rater | ICC (95% CI) |
| --- | --- |
| 1 | 0.997 (0.560, 1.00) |
| 2 | 0.998 (0.923, 1.00) |
| 3 | 1.00 |
| 4 | 0.999 (0.975, 1.00) |
| 5 | 0.922 (-0.092, 1.00) |
| 6 | 0.999 (0.973, 1.00) |
| 7 | 0.999 (0.548, 1.00) |
| 8 | 0.999 (0.230, 1.00) |
| 9 | 0.999 (0.977, 1.00) |
| 10 | 0.977 (0.906, 1.00) |
| 11 | 0.999 (0.548, 1.00) |
| 12 | 0.997 (0.578, 1.00) |
| 13 | 1.00 |
| 14 | 0.998 (0.912, 1.00) |
| 16 | 0.999 (0.548, 1.00) |
| 17 | 0.992 (0.209, 1.00) |
| 18 | 0.999 (0.978, 1.00) |
| 19 | 1.00 |
| 20 | 1.00 |
| 21 | 0.999 (0.975, 1.00) |
| 22 | 0.999 (0.174, 1.00) |

ICC: Intra-class Correlation Co-efficient; CI: Confidence Intervals.

does not represent how the physiotherapist would conduct an assessment in person. This is a limitation of this study and any interpretation of these results should take this into account. The raters within this study were highly experienced Neuromuscular Physiotherapists and were all active participants within a specialist national network (SMA REACH UK) which involves regular training and updates to improve clinical practice. Therefore, it could be assumed their clinical skills in undertaking this test with a patient would be sufficient. Furthermore, it would not have been feasible or indeed ethical to ask over 20 physiotherapists to assess the same patient(s) in this study due to the issues surrounding fatigue and burden for the patient. To overcome this limitation and ensure quality of RHS testing technique in the future, North Star/SMA REACH UK network physiotherapists could be asked to video an assessment which would then be reviewed for quality assurance purposes by the SMA REACH UK team, replicating the approach used in clinical trials.

This study has demonstrated that national reliability testing within the SMA REACH UK neuromuscular network can be conducted virtually. This further supports the function of the SMA REACH UK network in ensuring the UK is clinical trial ready. SMA REACH UK is co-ordinating the UK's implementation of the nusinersen managed access agreement (MAA). This study has demonstrated the inter-and intra-rater reliability of the RHS (an end-point of treatment efficacy in the MAA) when used by physiotherapists within this network and has also demonstrated the network is effective in delivering training both in person and virtually. The raters in this study were extremely experienced physiotherapists with median 9.5 years' experience in neuromuscular conditions, therefore caution should be applied in generalising results to less experienced physiotherapists.

This study has, for the first time, described reliability and agreement separately for patients with type 2 and 3 SMA. They are distinctly different phenotypes, and the results demonstrate high reliability and agreement for both types of SMA using this psychometrically robust scale.

Although recommended as the statistical test of choice to assess reliability by the FDA [38] the ICC value in absence of clinical context can easily be mis-interpreted. Kottner and Streiner [27] highlight reliability properties and agreement as distinctly separate concepts. The ICC is a ratio concerned with variability of scores, and agreement is the degree to which measures differ/agree, with the latter being more straightforward to interpret and grounded in clinical sensibility. This study is the first study to look at both reliability and agreement properties of an SMA functional scale. Bland Altman analysis has not been employed previously to assess the agreement of outcome measures in SMA. This form of analysis provides greater understanding of the scale with regards agreement in relation to test scoring (precision).

This study is transparent regarding clinical meaningfulness and interpretation of inter and intra-rater reliability of the RHS due to providing raters raw scores (Tables 1 and 2, Fig 1) and the expert set limits of agreement (precision) which these are compared against (Figs 2 and 3). This study has identified the inter and intra-rater measurement error/precision of the RHS, when used by UK physiotherapists within the SMA REACH UK network, is conservatively ±2 points. Therefore, observed changes in RHS scores between evaluations that lie within ±2 points should be interpreted with caution as they may not represent clinical change and rather reflect the reliability of the rater's scoring. In cases where a physiotherapist has measured a difference in ability of ±3 points this is unlikely to be due to measurement error. It has not been within the scope of this study to investigate the natural history of change within patient over time using the RHS, further investigations regarding longitudinal natural history in SMA 2 and 3 using the RHS are currently in progress.

The RHS has high inter and intra-rater reliability and agreement when being used by experienced neuromuscular physiotherapists from the North Star/SMA REACH UK network.

## Conclusions

This is the first study to report upon the inter and intra-rater reliability properties of the RHS. It has demonstrated the RHS has high inter and intra-rater reliability from a statistical perspective and anchors this to the clinical interpretation of agreement (precision of between/within raters scoring) as ±2 points for both inter and intra-rater reliability. The virtual approach of conducting the reliability testing nationally achieved a high response rate, was cost effective and could be repeated easily again in the future. Whilst the reliability of the RHS has been demonstrated in a UK cohort of experienced neuromuscular physiotherapists further work is required to determine the minimally clinically important difference of the RHS, test-retest reliability of the scale, and change over time regarding natural history and longitudinal trajectories.

## Supporting information

**S1 Dataset. Study minimal dataset.** This file contains the minimal dataset underpinning the results for this study. Demographic data is not included in this minimal dataset due to the risk of indirectly identifying participants. Please contact Dr Salma Samsuddin, SMA REACH UK & ISMAC UK Trial Manager via salma.samsuddin@gosh.nhs.uk for any queries regarding data access.
(XLSX)

## Acknowledgments

Expert Physiotherapists involved in RHS development–Anna Mayhew, Marion Main, Elena Mazzone, Jacqueline Montes

North Star Network/SMA REACH UK physiotherapists

SMA REACH UK network:

Dr Francesco Muntoni (Chief investigator), Great Ormond Street Hospital & UCL Great Ormond Street institute of Child Health

Dr Anna Mayhew (Collaborator site), Dr Volker Straub, Dr Chiara Marini-Bettolo, Institute of Genetic Medicine, Newcastle University & The Newcastle upon Tyne Hospitals NHS Foundation Trust

Dr Deepak Parasuraman, Birmingham Heartlands Hospital, University Hospitals Birmingham NHS Foundation Trust

Dr Anirban Majumdar, Dr Kayal Vijayakumar, Bristol Royal Hospital for Children, University Hospitals Bristol NHS Foundation Trust

Dr Iain Horrocks, Royal Hospital for Children, NHS Greater Glasgow & Clyde

Dr Anne-Marie Childs, Leeds Teaching Hospitals NHS Trust

Dr Stefan Spinty, Alder Hey Children's NHS Foundation Trust

Dr Elizabeth Wraige, Dr Vasantha Gowda, Evelina London Children's Hospital, Guys & St Thomas's NHS Foundation Trust

Dr Imelda Hughes, Royal Manchester Children's Hospital, Manchester University NHS Foundation Trust

Dr Gabby Chow, Nottingham University Hospitals NHS Trust

Professor Tracey Willis, The Robert Jones and Agnes Hunt Orthopaedic Hospital NHS Foundation Trust

Dr Sithara Ramdas, Oxford Children's Hospital, Oxford University Hospitals NHS Foundation Trust

Dr Christian deGoede, Royal Preston Hospital, Lancashire Teaching Hospitals NHS Foundation Trust

Dr Min Ong, Sheffield Children's NHS Foundation Trust

Dr Marjorie Illingworth, Southampton General Hospital, University Hospital Southampton NHS Foundation Trust

Dr Nahim Hussain, Leicester Royal Infirmary, University Hospitals of Leicester NHS Trust

Dr Elma Stephens, Royal Aberdeen Children's Hospital, NHS Grampian

Dr Deepa Krishnakumar, Addenbrooke's Hospital, Cambridge University Hospitals NHS Foundation Trust

## Author Contributions

**Conceptualization:** Danielle Ramsey, Gita Ramdharry, Mariacristina Scoto, Francesco Muntoni, Amanda Wallace.

**Data curation:** Danielle Ramsey.

**Formal analysis:** Danielle Ramsey.

**Funding acquisition:** Francesco Muntoni.

**Investigation:** Danielle Ramsey.

**Methodology:** Danielle Ramsey, Gita Ramdharry, Amanda Wallace.

**Project administration:** Danielle Ramsey, Mariacristina Scoto.

**Supervision:** Gita Ramdharry, Mariacristina Scoto, Amanda Wallace.

**Visualization:** Danielle Ramsey.

**Writing – original draft:** Danielle Ramsey.

**Writing – review & editing:** Danielle Ramsey, Gita Ramdharry, Mariacristina Scoto, Francesco Muntoni, Amanda Wallace.

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
