## [Decision Letter · Decision Letter 0]

16 Jun 2022

PONE-D-21-36288Revised Hammersmith Scale for Spinal Muscular Atrophy: inter and intra-rater reliability and agreementPLOS ONE

Dear Dr. Ramsey,

Thank you for submitting your manuscript to PLOS ONE. After careful consideration, we feel that it has merit but does not fully meet PLOS ONE’s publication criteria as it currently stands. Therefore, we invite you to submit a revised version of the manuscript that addresses the points raised during the review process.

Please note that we have only been able to secure a single reviewer to assess your manuscript. We are issuing a decision on your manuscript at this point to prevent further delays in the evaluation of your manuscript. Please be aware that the editor who handles your revised manuscript might find it necessary to invite additional reviewers to assess this work once the revised manuscript is submitted. However, we will aim to proceed on the basis of this single review if possible.  Your manuscript has been assessed by an expert reviewer, whose comments are appended below. The reviewer has highlighted concerns about some aspects of the methodology and statistical analysis. Please ensure you respond to each point carefully in your response to reviewers document, and modify your manuscript accordingly.

We look forward to receiving your revised manuscript.

Kind regards,

Joseph Donlan

Editorial Office

PLOS ONE

Journal Requirements:

4. Thank you for stating the following in the Financial Disclosure section: "This study was supported, in the UK, by the SMA REACH UK project (www.smareachuk.org). FM is the chief investigator of the SMA Reach project. Commercial funding for the SMA REACH UK project is provided by Biogen Inc (REC reference: 13/LO/1748, IRAS project ID: 122521), via UCL and GOSH. Historically, funding of the SMA REACH UK Project has also been provided by the SMA Trust and Muscular Dystrophy UK (07DN02; 37787 http://www.musculardystrophyuk.org/grants/clinical-trial-coordinators/), the MRC Translational Research Centre at UCL and Newcastle (MR/K501074/1), and the National Institute for Health Research Biomedical Research Centre (515048) at Great Ormond Street Hospital for Children NHS Foundation Trust and University College London  (http://www.gosh.nhs.uk/research-and-innovation/nihr-great-ormond-street-brc/about-brc). The funders had no role in study design, data collection and analysis, decision to publish, or preparation of the manuscript."

We note that you received funding from a commercial source: Biogen Inc

6. Please note that in order to use the direct billing option the corresponding author must be affiliated with the chosen institute. Please either amend your manuscript to change the affiliation or corresponding author, or email us at plosone@plos.org with a request to remove this option.

7. Please upload a new copy of Figures 1-4 as the detail is not clear. Please follow the link for more information: https://blogs.plos.org/plos/2019/06/looking-good-tips-for-creating-your-plos-figures-graphics/" https://blogs.plos.org/plos/2019/06/looking-good-tips-for-creating-your-plos-figures-graphics/

8. We note that Figure 1 in your submission contain copyrighted images. All PLOS content is published under the Creative Commons Attribution License (CC BY 4.0), which means that the manuscript, images, and Supporting Information files will be freely available online, and any third party is permitted to access, download, copy, distribute, and use these materials in any way, even commercially, with proper attribution. For more information, see our copyright guidelines: http://journals.plos.org/plosone/s/licenses-and-copyright.

Reviewers' comments:

Reviewer's Responses to Questions

**Comments to the Author**

1. Is the manuscript technically sound, and do the data support the conclusions?

Reviewer #1: Yes

2. Has the statistical analysis been performed appropriately and rigorously? 

Reviewer #1: Yes

3. Have the authors made all data underlying the findings in their manuscript fully available?

Reviewer #1: Yes

4. Is the manuscript presented in an intelligible fashion and written in standard English?

Reviewer #1: Yes

5. Review Comments to the Author

Reviewer #1: The authors have been rigorous in assessing the reliability of the HMS. On my part, I find few observations, but these observations may be fundamental. In particular, I think the study may be a good marker for further psychometric studies of the HMS.

1. The assessment of the reliability of the score first assumes that the unidimensionality of the items included in this score is satisfactory. Even in a small sample, the authors should report evidence of unidimensionality of the instrument, even with "simple" statistics, e.g., inter-item correlation, item-test correlation, etc.

2. In Table 1, they can also report the standard deviation or other univariate estimator of score variability.

3. "...were consulted regarding setting the level 190 of agreement for the Bland Altman analysis [17]. The experts agreed that the limits of agreement (precision) of the anticipated differences between raters/intra-rater for the total score of the test would lie within ±2 points overall and therefore ±2 was the set limits of agreement (precision) for this study..."

Here it is required to describe what was the rationale for arriving at this magnitude of anticipated differences.

4. It is advisable to summarily describe the content of the HMS items, perhaps in a section within the Method, e.g., Instrument.

5. I suggest including graphs of the distribution of scores, e.g., histograms.

6. The decision to use a two-week time interval may be a bit more applied.

7. In this first reliability study, internal consistency should also be reported, for example with coefficient alpha (with confidence intervals).

6. PLOS authors have the option to publish the peer review history of their article (what does this mean?). If published, this will include your full peer review and any attached files.

Reviewer #1: **Yes: **César Merino-Soto

---

## [Author Response · Author response to Decision Letter 0]

25 Sep 2022

Dear Mr Domini,

We thank you for the careful reading of our manuscript.

Please find attached our second revision of the manuscript PONE-D-21-36288 “Revised Hammersmith Scale for Spinal Muscular Atrophy: inter and intra-rater reliability and agreement" for further consideration for publication in PLOS ONE. 

We have addressed the point in this letter & the manuscript related to journal requirements to:

‘include a non-author institutional contact for data access in the interest of maintaining long-term data accessibility. At this time, please provide a non-author point of contact that is abe to receive queries regarding data access’ 

Author’s response:

As requested we have attached a clean and updated version of the manuscript, a tracked changes version of the manuscript which we consider would now meet your required specifications. In the manuscript we have added detail on page 24, line 542-543 to contact “Dr Salma Samsuddin, SMA REACH UK & ISMAC UK Trial Manager via salma.samsuddin@gosh.ac.uk for any queries regarding data access.”

We have also further updated the data availability statement to include the same information.

Updated data availability statement:

“The minimal dataset underpinning the results of this study can be found in the supporting information S1 Dataset. This does not include detail regarding the demographics of the raters included in the study, this information, their affiliation with the SMA REACH UK sites and small population studied would mean even anonymised data would have several indirect identifiers which may risk breaking anonymity. Minimal information pertaining to the raters can be found within the participants section within results. The minimal dataset can also be found within the results section in tables 1, 2 and 3 and within Figures 1, 2 and 3. 

Data beyond the minimal dataset required for this study are not publicly accessible due to it containing sensitive data pertaining to human research participants and may compromise anonymity. However, external parties can request data in aggregate form from the SMA REACH UK steering committee for researchers who meet the criteria for access to confidential data. Please contact Dr Salma Samsuddin, SMA REACH UK & ISMAC UK Trial Manager via salma.samsuddin@gosh.nhs.uk for any queries regarding data access.”

We hope this revised version will now be acceptable for publication in the PLOS ONE. If you require any additional information, please do not hesitate to contact us.

Yours sincerely,

Danielle Ramsey

---

## [Decision Letter · Decision Letter 1]

29 Nov 2022

Revised Hammersmith Scale for Spinal Muscular Atrophy: inter and intra-rater reliability and agreement

PONE-D-21-36288R1

Dear Dr. Danielle Ramsey

We’re pleased to inform you that your manuscript has been judged scientifically suitable for publication and will be formally accepted for publication once it meets all outstanding technical requirements.

Kind regards,

Jae-Young Hong

Academic Editor

PLOS ONE

Reviewers' comments:

Reviewer's Responses to Questions

**Comments to the Author**

1. If the authors have adequately addressed your comments raised in a previous round of review and you feel that this manuscript is now acceptable for publication, you may indicate that here to bypass the “Comments to the Author” section, enter your conflict of interest statement in the “Confidential to Editor” section, and submit your "Accept" recommendation.

Reviewer #1: All comments have been addressed

2. Is the manuscript technically sound, and do the data support the conclusions?

Reviewer #1: Yes

3. Has the statistical analysis been performed appropriately and rigorously? 

Reviewer #1: Yes

4. Have the authors made all data underlying the findings in their manuscript fully available?

Reviewer #1: Yes

5. Is the manuscript presented in an intelligible fashion and written in standard English?

Reviewer #1: Yes

6. Review Comments to the Author

Reviewer #1: The authors have effectively addressed each point made in the review. I am almost completely satisfied with the modifications and justifications made.

I state "almost" because I would still like to insist that the authors report some concurrent evidence with their data on item-test relationship and internal consistency reliability. Although the authors have supported these properties based on a previous study (reference 17) with Rasch modeling results, this is not necessarily replicable in other contexts and samples, even more so in small samples. The authors induce validity from another study, but do not corroborate it in the present sample (please see references to the problem below).

Overall, this observation does not create a substantial limit to the publication of the manuscript. Also, pointing out these references does not compel the authors to cite them.

References

https://www.reumatologiaclinica.org/en-metric-studies-compliance-questionnaire-on-articulo-S2173574321001660

https://www.elsevier.es/es-revista-revista-colombiana-reumatologia-374-estadisticas-S0121812320300566

7. PLOS authors have the option to publish the peer review history of their article (what does this mean?). If published, this will include your full peer review and any attached files.

Reviewer #1: **Yes: **Cesar Merino-Soto

---

## [Editor Report · Acceptance letter]

12 Dec 2022

PONE-D-21-36288R1 

Revised Hammersmith Scale for Spinal Muscular Atrophy: inter and intra-rater reliability and agreement 

Dear Dr. Ramsey:

I'm pleased to inform you that your manuscript has been deemed suitable for publication in PLOS ONE. Congratulations! Your manuscript is now with our production department. 

Kind regards, 

on behalf of

Professor Jae-Young Hong 

Academic Editor

PLOS ONE